# A Study Protocol for Occupational Rehabilitation in Multiple Sclerosis

**DOI:** 10.3390/s21248436

**Published:** 2021-12-17

**Authors:** Marco Trombini, Federica Ferraro, Giulia Iaconi, Lucilla Vestito, Fabio Bandini, Laura Mori, Carlo Trompetto, Silvana Dellepiane

**Affiliations:** 1Department of Electrical, Electronics and Telecommunication Engineering and Naval Architecture (DITEN), Università degli Studi di Genova, Via all’Opera Pia 11a, I-16145 Genoa, Italy; marco.trombini@edu.unige.it (M.T.); federica.ferraro@edu.unige.it (F.F.); giuliaiac24@gmail.com (G.I.); 2Department of Neurosciences, Rehabilitation, Ophthalmology, Genetics, and Maternal and Children’s Sciences (DINOGMI), Università degli Studi di Genova, Largo Paolo Daneo 3, I-16132 Genoa, Italy; lucillavestito79@gmail.com (L.V.); laura.mori@unige.it (L.M.); ctrompetto@neurologia.unige.it (C.T.); 3Ospedale Policlinico San Martino IRCCS, Largo Rosanna Benzi 10, I-16132 Genoa, Italy; 4Struttura Complessa di Neurologia-Ospedale Villa Scassi ASL 3, Corso Onofrio Scassi 1, I-16149 Genoa, Italy; fabio.bandini@asl3.liguria.it

**Keywords:** Internet of Medical Things, telerehabilitation, multiple sclerosis, remote patient monitoring, occupational rehabilitation

## Abstract

Digital medical solutions can be very helpful in restorative neurology, as they allow the patients to practice their rehabilitation activities remotely. This work discloses ReMoVES, an IoMT system providing telemedicine services, in the context of Multiple Sclerosis rehabilitation, within the frame of the project STORMS. A rehabilitative protocol of exercises can be provided as ReMoVES services and integrated into the Individual Rehabilitation Project as designed by a remote multidimensional medical team. In the present manuscript, the first phase of the study is described, including the definition of the needs to be addressed, the employed technology, the design and the development of the exergames, and the possible practical/professional and academic consequences. The STORMS project has been implemented with the aim to act as a starting point for the development of digital telerehabilitation solutions that support Multiple Sclerosis patients, improving their living conditions. This paper introduces a study protocol and it addresses pre-clinical research needs, where system issues can be studied and better understood how they might be addressed. It also includes tools to favor remote patient monitoring and to support the clinical staff.

## 1. Introduction

Multiple sclerosis (MS) is a common neurological disease that affects the central nervous system (CNS) [1] and provokes the impairment of several functions including motor skills [2], balance [3], cognition [4], and activities of daily living (ADLs). Cognitive impairment (CI) affects up to 70% of the MS population and refers to domain-specific deficits rather than uniform global cognitive decline [5]. The patient with MS (pwMS) may encounter difficulties in information processing speed, sustained and selective attention, learning and episodic memory, with impaired executive functions in the more advanced progressive stages [5], as well as visuospatial problems [6], which negatively affect social, occupational activities, and the quality of life in general.

As it is well known, therapy for MS can be divided into two categories: disease-modifying therapies (DMT) and symptomatic or supportive therapies. In principle, DMTs might improve cognition as their agents are primarily designed to arrest the disease and prevent relapses, but the actual benefit on improving cognition is still matter of debate [7]. On the other hand, the goal of cognitive and behavioral rehabilitation strategies is to enhance patients’ ability as related to executive functional tasks.

Although the focus on this topic is a relatively recent phenomenon, the growth of research studies addressing the need for effective cognitive rehabilitation programs has been substantial over the past decade, as is well documented in the literature [8,9,10,11,12,13,14,15,16,17,18,19]. In this context, for instance, in recent years the Italian National MS Society (Associazione Italiana Sclerosi Multipla-AISM) has recommended remedial interventions and accommodations to manage cognitive impairment and improve everyday functioning in both adult and pediatric MS populations [20]. Rehabilitation therapy is considered an interdisciplinary approach in which different professionals carry out a rehabilitation intervention aimed at the individual patient, called the Individual Rehabilitation Project (PRI), in order to improve their physical, psychological, and social functions and maintain their autonomy. The exploitation of novel technologies, such as Virtual Reality (VR) and Exergame, is suggested as a supplement to the rehabilitation therapy of MS patients [21,22].

Thanks to the impressive development of the last few decades in Information and Communication Technologies (ICT), including Electronics, Telecommunications, and Signal Processing, the Internet of Medical Things (IoMT) is starting to represent a preferred solution with the goal of supporting rehabilitation in a continuous and safe way, guaranteeing both social distancing and the reduction of travel to the rehabilitation site.

In this context, the present article describes the “Solution Towards Occupational Rehabilitation in Multiple Sclerosis (STORMS)” project as an example of an IoMT work in progress and study protocol devoted to remote hospital and home monitoring of patients with multiple sclerosis. STORMS is one of the two winners of the “2020 Digital Innovation in Multiple Sclerosis” award, sponsored by Merck [23], whose goal is to promote adaptation and coexistence with the disease through new digital technologies.

With the aim of improving the patient’s quality of life and increasing the opportunities for contact with the multidimensional medical team, STORMS is based on the ReMoVES IoT system developed at the University of Genoa and described in previous works [24]. Designed to provide remote tests for the assessment of patients’ impairments and to provide exercises to support motor/cognitive rehabilitation, ReMoVES has already been tested on neurological [25,26,27] and rheumatic patients [28] as well as on the elderly [29].

ReMoVES interface is very simple and intuitive even for people with impairments, the delivery of the exercises is personalized and follows the individual activity plan defined by the therapist in agreement with a multidimensional team.

Based on various devices and motion sensors (Touchscreen, Kinect, Leap Motion) ReMoVES Patient Client performs calibration on the individual user in order to provide “tailored” results and allows the therapist to monitor tests and activities through a Therapist Client that can be consulted from different technologies (PCs, phones, tablets). On the server side, all activity-related data and signals are processed using adaptive approaches, enabling performance quality assessment, detection of the pattern of complex activities actually performed, and detection of any compensating movements. The extraction of significant data is performed only after suitable nonlinear filters and adaptive segmentation which allow to extract statistics and key indicators from signal fragments characterized by stationarity properties and related to homogeneous measurement states.

The playful component of the exergame will help to increase motivation and adherence to rehabilitation. At the same time, the system will favor the practice of task-oriented exercises and muscle strengthening, favoring the improvement of the patient’s functionality. To take into account the wide variability of patients’ conditions and disease progression, the most appropriate exercises can be assigned and the complexity of the required task can be adapted by defining a range of difficulty levels.

Thanks to the collection, integration and remote analysis of patient signals and data, the solution allows continuous monitoring of the activities on which the therapist constantly updates the personalized exercise plan. According to the ISO/IEC/IEEE 12207 standard, the development of the ReMoVES IoT system followed the requirements engineering process consisting in defining the scope of the project, the IoT system and its requirements [30], based on the defined stakeholders and to the final destination user group.

Based on such a past experience, new stakeholder roles have been defined in close coordination between medical and technical developers to address the new MS use case target and update related goals and actions. Based on the clinical needs of MS patients and the particular types of CI they are affected by, new activities delivered in form of exergames have been developed to address functional abilities of attention, memory, executive functions, and information processing speed.

The main originalities and strengths of this work are as follows:Implementation of an IoMT system for the assessment and support of in-hospital and in-home rehabilitation in people with MS.Exploitation and rapid adaptation of the ReMoVES IoT system to the new target user group.Ability to exercise, monitor, evaluate and analyze both the motor aspects (upper limbs, lower limbs, trunk movement, and balance control) and cognitive aspects (attention, memory, working memory, etc.).Provision of a personalized service tailored to the needs of the individual who has been assigned an individual care plan.Ease of use, low cost, and integrability with other systems.Robustness and resilience with respect to temporary telecommunication problems.Adaptive nonlinear filtering and segmentation of signals for data extraction, analysis, and visualization.

The present manuscript is structured as follows. The state-of-the-art is presented in Section 2, focusing on cognitive impairments on MS patients and digital rehabilitative approaches to MS treatment, highlighting the main differences with ReMoVES. Section 3 gives a brief introduction to ReMoVES and the deployed technologies. Furthermore, the criteria of the participating population are here defined along with the operational protocol. In Section 4, the developed activities are described in depth including the cognitive functions that are stimulated. In Section 5, possible future analysis of the results are discussed. In the end, a discussion and conclusion on the present work are given in Section 6, also providing a glimpse into its future developments.

## 2. State-Of-The-Art

In general, signs of cognitive involvement are present already in early stages of the disease process [11], even though severe cognitive impairment is more likely in persons with secondary progressive MS. Indeed, approximately half of persons with MS report having either minimal or mild cognitive difficulties within the first year of diagnosis [31], with greater complaints over the first decade. Furthermore, although uncommon, some persons with MS present cognitive impairment as their primary symptom. In addition, sometimes cognitive issues may be present pre-clinically [31].

As a matter of fact, these impairments yield to major consequences for everyday life. Indeed, CI is the leading cause of occupational disability and of difficulties in ADLs for such patients [32].

To manage such impairments, some recommendations include more comprehensive assessment for anyone who tests positive for cognitive impairment on cognitive screening or demonstrates substantial cognitive decline, as well as neuropsychological evaluation for any unexplained change in academic performance or behavioral functioning in school-aged children with MS. Evidence suggests that cognitive rehabilitation has a long-term impact well beyond the treatment period and might enhance cognition in the face of future brain changes. Such sustained effects have been documented in the literature on aging, in which cognitive rehabilitation not only improved everyday life activities, but also resulted in a 29% reduction in dementia risk 10 years after treatment [33].

People with MS also belong to the so-called “fragile” populations subjects who are at higher risk in relation to pandemic emergencies, such as the current one. ICT and IoT solutions can represent a very important tool for supporting rehabilitation and favoring adaptation and coexistence with disability.

### 2.1. Technologies Employed in Multiple Sclerosis

The importance of using technology in the treatment of MS has long been acknowledged, so that several solutions addressing diagnosis, monitoring, and rehabilitation can be found in the literature [34,35]. Several studies have shown that patients better perceive their goals and physical and mental well-being, thanks to the improved feedback that technology provides, thus leading to a better practice and to an enhanced engagement in the therapy [36]. In addition, technology support also favors intense and repetitive training that yields effective results in functional recovery for MS patients [21,37]. Of note, the IoMT technologies can support patients in taking control of their own MS disease and collaborate more effectively with the clinical staff [38]. Despite the large interest towards assistive technology in MS, solutions are still not as widespread as they may be, because of some barriers for patients in terms of usability and feasibility, and also because of the high costs of some devices [39]. Indeed, MS patients can experience difficulties in dealing with technological devices, as well as poor skills in using it [40]. In addition, technological solutions made up of wearable devices or controllers may require external support from caregivers, limiting the independent use. Furthermore, the high cost of solutions hinders the possibility of a large home-based usage.

### 2.2. Exergames in the Field of Rehabilitation

The term exergame refers to video games that impart physical exercise/support rehabilitation practice (in the context of their clinical application) in which the repetitive and task-oriented components of rehabilitation activities are reformulated in terms of video game tasks. In recent times, exergames have gained great popularity and demonstrated scientific reliability, thus going beyond their original goal of mere entertainment. The exergames can also be considered as a VR tool, which can be a safe instrument to access activities otherwise not accessible to the person with cognitive and motor disabilities in everyday life contexts. Furthermore, gamification [41] determines a motivating and engaging environment in order to contrast boredom and fatigue in patients, and thus fostering the continuity of care.

Exergames have demonstrated their utility for both cognitive and motor rehabilitation, as they exert beneficial effects on attention, visuo-spatial function, executive control, strategic planning, and processing speed [42,43]. Several studies reported the efficacy in maintenance and improvement of cognitive functions in the elderly population [44] and in post-stroke rehabilitation [45]. Some solutions providing exergames for entertainment have been utilized for rehabilitative purposes, also in the context of MS patients [22,46,47].

### 2.3. Differences with Removes System

Some rehabilitation protocols for multiple sclerosis disease have been presented in the literature. However, assessment and rehabilitation are often just [48] cognitive or [49] physical. On the contrary, in the present study, the multidimensional team creates individual rehabilitation plans aimed at reintegrating cognitive and motor aspects that can be studied through the analysis of data obtained from motion sensors such as Microsoft Kinect, recognized as a tool of great validity and efficacy for the study and analysis of human motility [50]. In addition, it is an easily usable and convenient device. Having additional components could lead to a physical encumbrance problem and therefore limit not only the installation of the implant directly at the patient’s home, but also the rehabilitation plan itself.

The particular attention paid to signal processing is another important aspect in the description of the proposed solution. In general, existing systems do not pay attention to the lack of stationarity and homogeneity of the acquired signal, resulting in erroneous or insignificant statistical analysis results. To cope with this drawback, adaptive signal segmentation was introduced at each game session in order to separate the complex pattern of the patient’s execution task into primitive elements. The studies in the literature have sections dedicated to the analysis of the results obtained with their systems, even commercial, i.e., Jintronix and Mira, but there is lack of the preprocessing phase of the raw signal [51,52].

In addition, ReMoVES is programmed to acquire the positions of all the patient’s joints while they carry out their rehabilitation program. The performance characteristics such as angles, range of motion, and trajectories will then be identified in order not only to assess the correctness of the movement suggested by the activity in question, but also to analyze any compensations and strategies.

Due to the large individual variability between patients and their impairments, and given the relatively small sample of users, a supervised approach or generalized model is not practical in this case. Compared to other systems based on machine learning techniques (such as in [53]), a wide use of unsupervised adaptive analysis methods is used here to derive useful information from the acquired signals.

## 3. Materials and Methods

In this section, after a description of the ReMoVES system and the technologies used, the inclusion criteria, operating protocol, and patient–client activation procedure are introduced.

### 3.1. The Removes System

ReMoVES is an IoT system for remote rehabilitation developed at the Department of Electrical, Electronic and Telecommunications Engineering and Naval Architecture (DITEN) of the University of Genoa. An ad hoc version of the ReMoVES services was designed for the STORMS project and a series of exergames suitable for MS patients were created.

According to the IEEE Standard for an Architecture Framework for the Internet of Things [54], Figure 1 shows the architecture of the developers point of view. The close collaboration between the multidimensional team and the technical research institute allows the definition of targeted activities, as in this case for pwMS, identifying both exercises (cognitive, motor, visual, balance tasks) and their evaluation (values and graphs of the trend of the task sessions). Following the standard engineering process, the Therapist Client is then built according to the target user group requirements and the needs of the clinical staff.

Figure 2 shows the patient–viewpoint architecture. On the basis of the patient’s observations, the multidimensional team updates the individual care plan consisting of an appropriate program of prescribed activities. Through the cloud, the Patient Client receives the assigned scheduling of exergames, either in the hospital or at the patient’s home. All data and signals are collected and accessible to clinical staff through the Therapist Client to acquire feedback on activity performance.

From the IoT viewpoint, ReMoVES has a classical four-layer architecture, as described in detail in [29], which is briefly summarized in the following.

Referring to Figure 3, the physical layer describes the so-called “Things” as defined in [54]. The STORMS project uses the Microsoft Kinect V2 for motor/cognitive activities.

Kinect is a motion-sensitive input device that produces a depth map of the environment using a time-of-flight camera. It offers a 70° × 60° field of view and recognition up to 4.5 m away. The Microsoft Kinect sensor can simultaneously 3D track up to 25 key joints of the framed human body. The 3D position signals are sampled at a frequency of 10 Hz. Several studies have shown that Microsoft Kinect V2 can validly obtain spatio-temporal parameters and also be an acceptable tool for rehabilitation due to its low cost and adequate spatial accuracy (with an order of magnitude of centimeters) [55].

Automatic execution of the scheduled task, synchronization with sensors, and acquisition and storage of signals and data are performed via a standard PC when patient is in the hospital (where an additional patient identification phase is required) or via an industrial PC without a keyboard when run at home.

Exergames are digital games that encourage patient to carry out motor/cognitive exercises. A detailed description of the exergames for pwMS is presented in Section 4.

Unity is a cross-platform game engine used for the development of 3D and 2D games as well as virtual reality applications [56]. Among the programming languages supported by Unity, C# was chosen for developing the ReMoVES exergames. The interface configuration between Microsoft Kinect V2 and the Unity engine is straightforward as the manufacturer provides the Software Development Kit (SDK) and an add-on for Unity. Developers can easily access the positions and the orientations of body joints for direct use in the virtual environment of the game scenes. In the ReMoVES project, all the 2D and 3D graphical resources have been downloaded from different online sources with a Creative Commons license.

The network layer represents the link between physical and server layers. To obtain a resilient functionality facing eventual connection disruption, data log-files are temporarily stored in the local unit in JavaScript Object Notation (JSON), and the actual transmission is executed when connection is available via Ethernet or Wi-Fi.

Data communication is in secure mode based on hypertext transfer protocol secure (HTTPS), the communication protocol is encrypted using transport layer security (TLS), and certificates are issued by the Let’s Encrypt authority.

The server layer can manage content-independent dataflow to be compliant with software reuse logic. Server software consists of a traditional Linux–Apache–MySQL–PHP (LAMP) stack, and provides data storage solutions, data processing methods, and a web application for clinicians to view information through dedicated graphic interfaces. Only three types of application programming interfaces (APIs) for the management of client/server data synchronization are used.

The MySQL relational database contains a structured collection of JSON files, each of them describing an array of temporal events. In each element of the array, there are key-value pairs that provide data. Some keys are common to all exergames, in addition, other keys are provided depending on the game.

Finally, the application layer includes the therapist client, who provides the medical staff with all the necessary information on the patients being followed, displaying their performance in graphical mode and verifying their current treatment plan.

### 3.2. Inclusion and Exclusion Criteria

For this study, pwMS from the Neurorehabilitation Clinic of the San Martino-IRCCS Hospital in Genoa will be recruited. They must be self-sufficient and can be recruited regardless of disease subtype, gender and age. The inclusion criteria are as follows:Clinically defined MS with Expanded Disability Status Scale (EDSS) < 6.Absence of significant linguistic-communicative deficits (Token test score > 25).Normality of the mental state i.e., Mini-Mental State Examination (MMSE) > 24.Absence of psychiatric pathologies.Visual acuity not lower than 6/10.

Conversely, the exclusion criteria are as follows:Inability to maintain adequate visual fixation (for example, in case of nystagmus).Presence of post-chiasmatic perimetric defects.Photosensitive epilepsy.Poor compliance or, more generally, insufficient motivation to follow a continuous daily treatment such as the one in question.

Recruited patients will be evaluated using the following scales: Barthel Index (BI), Sollerman, Disability of arm, shoulder and hand (DASH), 10 m walk test (10MWT), 2 min walk test (2MWT), Berg Balance scale (BBS), Short Physical Performance Battery (SPPB), and 12-Item Short Form Survey (SF12) for the quality of life.

### 3.3. Operational Protocol

The operational procedure will be mostly carried out at home, in order to ensure support in dealing with the disease even in everyday life. Due to the user-friendly features of ReMoVES and the use of off-the-shelf components for the construction of the system, the current protocol could be replicated on a large scale.

The proposed treatment begins when patients make a first visit (time T0). Then, after an initial period of training in the hospital, the activity continues at home. The sessions last 30 min and are held daily for five days a week and for three weeks (the end term is defined T1 time).

Thereafter, patients are offered to continue the rehabilitation program at home for an additional four weeks, with the same schedule as before. Subjects will be re-evaluated after these additional 4 weeks of home treatment (time T2).

Patients are informed about the type of treatment and their written consent is always required.

### 3.4. Patient Client Activation Protocol

In the following, the phases necessary for the activation of the Patient Client are described.

The patient is identified through their personal ID.After the calibration phase, a simple video tutorial starts which explains the particular activity to be performed.The patient performs the activity, with a specific level of difficulty, according to the individual rehabilitation plan formulated and assigned by the multidimensional team.At the end of the game session, the patient receives feedback regarding the results of their performance.If an Internet connection is available, the client sends the session results to the server in order to process the data.

## 4. Activities for Occupational Rehabilitation

The present Section introduces the new exergames developed to meet the needs of the STORMS project, designed according to the result of the requirement analysis process. In particular, levels and functionalities are described. Then, the extracted data are described through the definition of the fields that appear in the log-files.

### 4.1. Exergames

The new set of exergames is mainly focused on cognitive recovery rather than motor rehabilitation. ReMoVES for the STORMS project includes six adaptive activities presented in visual format. The patients interact with the console through their body movements using the Microsoft Kinect sensor. Three types of movement, flexion–extension and abduction–adduction of the shoulder joint, and bilateral inclination of the trunk are monitored. An initial calibration phase and real-time adjustment adapt the game to the mobility of the user.

Each game is made up of simple 2D elements to prevent the patient from being distracted by irrelevant background elements. The performance during each session is recorded. At the end of the game session, the parameters useful for investigating the player’s performance are collected in a JSON file and provided for the analysis. Also motion data can be reconstructed, as the joints positions are recorded by the Kinect.

The JSON file is composed of an array that collects temporal events. Each element of the array expresses data in key-value form. Some keys such as *Time*, *Score* and *Kinect* (i.e., the position of the joints detected by the sensor) are common to all exergames, while *Level* refers to all games that have multiple levels. The features *CalibSide*, *RealSide* and *HandPositionX-Y* are present only in games that involve the movement of the arm, and they represent, respectively, the body side detected during calibration phase (0 for none, 1 for right side, 2 for left side), the hand detected during the game phase (0 for none, 1 for right hand, 2 for left hand), and the x and y positions of the hand in the game scene.

The main cognitive functions involved in games include attention, memory and executive functions. Among the sub-categories of such abilities, the games will mainly address the following:**Working memory**: describes the processes involved in the control, regulation, and active maintenance of information relevant to the task at the service of complex cognition.**Inhibition control**: it is an executive function that allows a person to control the impulses and behavioral responses to stimuli in order to choose the most appropriate behavior consistent with achieving the objectives.**Selective attention**: it is the act of focusing on a single object in the environment for a specific period of time.**Task switching and cognitive shifting**: they are two executive functions that involve the ability to, respectively, unconsciously or consciously shift the attention between one task and another.**Multitasking**: it is the ability to focus on multiple tasks or activities at once.**Sustained attention**: it is the ability to focus on an action or stimulation for an extended period of time.**Top-down attention task**: this type of attention refers to the ability to focus on specific features, objects or regions in space that are relevant to a goal, filtering out irrelevant stimuli.

The new cognitive games introduced here aim to treat some of the most common symptoms of multiple sclerosis such as coordination disorders, balance problems, and dizziness (Hot Air); vision disturbances which may also include impaired color vision (Owl’s Nest), cognitive disorders that incorporate problems with memory and learning, difficulties in maintaining concentration, difficulties in attention, in computational problems; and inability to perform operations of a certain complexity and in problems to correctly perceive the environment (Supermarket, Numbers, Shelf Cans, and Business by Car). It is possible to obtain indirect measurements of symptoms such as numbness of the body and/or extremities or spasticity which can complicate movement when games are performed with pelvic or limb movement.

The targeted cognitive domains of each exercise are summarized in Table 1.

In some exergames, the patient is encouraged to reach consecutive on-screen targets with the arm motion (reaching task). The more targets are taken, the higher the in-game score. Such games aim at improving hand–eye coordination, and spatial awareness. In addition, some other exergames promote the trunk balance used to guide an object along a path.

In the Owl Nest, Supermarket, Numbers, and Business By Car exercises there are different levels, from easy to extremely difficult, while Shelf Cans and Hot Air activities have a single level and can be used by patients to familiarize themselves with the system. This is in order to define the treatment plan based on the patients’ disability, aimed at selecting the most appropriate game and level to start and continue therapy.

In all the exergames, except Business By Car, the total game time is 90 s. Business By Car has a longer duration (600 s) to ensure that the patient can reach the end of the game even if he sometimes makes a mistake along the path.

### 4.2. Owl Nest

The goal is to grab the owls that randomly appear on the screen with the flexion–extension of the arm and bring them into a nest placed in the middle of the screen.

**First level:** no more than three owls can appear simultaneously, with no distracting elements. When the user brings an owl into the nest, another owl appears at a different point and the in-game score increases.**Second level:** some eagles appear on the screen as distracting elements. No more than five owls and three eagles can appear simultaneously. The time between the appearance of two consecutive eagles randomly ranges from 0 to 5 s. After 10 s the eagle disappears. Catching an eagle makes the game score decrease.**Third level:** the player is required to catch only the pink owls and do it as quickly as possible, since after 15 s the owls will disappear. Every time an owl disappears, another reappears in a different location until a maximum of seven owls are simultaneously on the game scene. The score decreases either when grabbing a blue owl or when a pink owl disappears.**Fourth level:** it is a combination of all the goals of the previous levels. No more than four owls and three eagles appear simultaneously. The user must bring the pink owls into the nest and avoid both eagles and blue owls. Eagles disappear after ten seconds, while owls disappear after 8 s.

Figure 4 shows some game levels while Table 2 shows the features extracted. The functions involved are coordination of voluntary limb movements, reaction time, inhibition and selective attention, processing speed, and hand–eye coordination.

### 4.3. Supermarket

This exergame is set into a supermarket where the player is instructed to buy some objects. A list of items to pick up is displayed at the game start. The user has a time between 8 and 25 s to memorize this list, depending on the level. When the game starts the patient has to take the correct objects by moving the arm. An audio feedback is provided, with either a positive or negative sound occurring in case of correct or incorrect action, respectively. In the first, third, and fourth levels, once the list has been completed and the patient collected all the objects, they reappear on the screen in changed position. The user can collect the stored objects again.

**First level:** a temporary list of three food names must be memorized in eight seconds. Then, the player has to collect the relative objects, moving them from the two lateral shelves into the shopping bag (placed in the middle of the screen). Some non-food distractors appear on the shelves. The semantic property (food or non-food) of the objects is the crucial correctness factor of the activity. The game score decreases if wrong objects are put into the bag. A visual feedback is also provided (initially yellow health bar turns red when mistakes are made).**Second level:** the player has to memorize four ordered objects in ten seconds. These objects must be collected among other distracting items sliding on the conveyor belt. The game score decreases if the wrong object is taken or if the order is not followed.**Third level:** the player must memorize and follow four sequenced instructions in twenty seconds. Each instruction refers to a different semantic characteristic (shape, color, or material) of the objects to be collected. Collected items will not re-appear.**Fourth level:** it is like the third level, but it differs for the higher number of the objects in the scene and the number of instructions (five instead of four, to remember in 25 s). In addition, the objects will reappear on the screen once collected.

In Figure 5, the screenshots of the four levels are provided, while Table 3 contains the list of features extracted. The aim of this game is to train memory (verbal, non-verbal, and visual), inhibition and selective attention, and hand–eye coordination.

### 4.4. Numbers

The patient has to pop some numbered (from 0 to 99) balloons according to temporary instructions. Four instructions alternate according to the level difficulty:pop the balloons in ascending order;pop the balloons in descending order;pop the balloons with even numbers;pop the balloons with odd numbers.

The number of balloons varies depending on the difficulty level; they also have different colors and sizes to make the game more dynamic and visually appealing. If the user takes a wrong balloon, a red mark appears at the bottom of the screen, otherwise a green one does. In both cases, all the balloons still on the screen will be destroyed and a new round will begin with new balloons.

**First level:** four balloons appear on screen. They must be popped in either ascending or descending order.**Second level:** it is as the first level, except that the patient is also required to pop either the even or the odd balloons and there are five balloons in the game scene. Two visuo-verbal stimuli are added. When the text relating to the assignment “take the odd numbers” shows, a red bird will appear for a few seconds flying from one side of the scene to the other. Conversely, a plane will appear on the screen for a few seconds when the text relating to the task “take the even numbers” appears.**Third level:** six balloons are simultaneously displayed. Once more, all the tasks can be performed, but this time the patient must remember the stimuli association previously described, because in the cases of “take the odd numbers” and “take the even numbers” no writing appears on the screen.**Fourth level:** it is structured like the third level, but also the writings “pop in ascending/descending order” will disappear after a couple of seconds.**Fifth level:** the player has to quickly pop as many correct balloons as possible before they disappear. In fact, all the four balloons will fly off and disappear from the screen. Only the tasks about the odd and the even numbers with their relative visuo-stimuli, will be displayed.

In Figure 6, the screenshots of some levels are provided. In Table 4, a short description of the parameters extracted is showed. The aim of this exergame is to improve the attention, the task and cognitive switching, the processing speed, and the hand–eye coordination.

### 4.5. Business by Car

In the *Business By Car* exergame, the patient drives a car along a randomly generated road. In particular, the car turns either left or right as the player moves the trunk laterally to the left or to the right, respectively. The increase of the car speed is progressive, until the player goes out of the carriageway, which causes a penalty in the score and returns the speed to the initial condition. In this last case the car will be re-positioned on the path. At the beginning of the game, a list of places to visit appears. The patient will have to memorize this list in a time that varies between ten and twenty seconds, based on the selected level. Afterwards, the game will start and the patient will have to drive the car along the path and select the correct street at the crossroads to pass by the required places. Once the errands are finished, a series of multiple choice questions will appear on the screen, relating to the list and places or about details present in the game scenes or in the buildings visited. To answer the questions the patient has to raise his arm and guide the hand that appears on the screen towards the answer button. In detail, the three levels will be described which differ according to the number of places to remember and the difficulty of the final questions.

**First level:** in the easiest level, the patient must remember only four places to visit. The buildings to see are simply written on the list, with no further writings, to make the goal clear. At a crossroads, the navigator at the bottom-right will indicate the correct route to take. If the patient happens to take the wrong path, a message will appear, reminding him/her of the correct place to visit and to pay more attention to the next crossroads. In the question scene, there will be two questions relating only to the stuff to do, e.g., "Was the first building to visit the Supermarket?"**Second level:** in this level the patient must remember five places. This time a real list of errands to be carried out will appear at the beginning of the game (e.g., "Do the grocery shopping at the Supermarket"). It can help the patient in providing context and remembering where he/she needs to go. The navigator does not indicate the correct path to take and only a warning message will appear on screen if the player goes on the wrong direction. The final questions are three, two of them about the list and the last one relating to a detail of a visited building (i.e., "What color was the Supermarket awning?").**Third level:** in the last level, the player must keep in mind six places. No warning message will appear if the wrong way is taken. In the final scene, the patient should answer to four questions, one related to the initial list and three concerning details seen in the visited buildings.

Figure 7 shows some screenshots of the exergame and its features are collected in Table 5. The aim is to train visual memory, attention, postural balance, and correction reactions.

### 4.6. Shelf Cans

In this exergame, the player is required to store items on a wall unit. In detail, the game activity is to move a colored can of soda towards the corresponding shelf. The up-right, up-left, and bottom-left corners are for orange, green, and red tin cans, respectively. Therefore, Shelf Cans contributes to improve the attentive capacities of patients, who must respect the rule of matching the color between the can of soda and the shelf correctly. The penalty for making a mistake is the reduction of the score accompanied by a disappointing sound. A screenshot of the game is provided in Figure 8 and Table 6 contains the list of the extracted parameters. This game is aimed at stimulating hand-eye coordination of voluntary arm movements, reaction time, inhibition and selective attention, and processing speed.

### 4.7. Hot Air

The game is set in a hilly landscape where a hot-air balloon flies and advances automatically. The player can control the lateral displacement of the balloon by performing trunk shifting. The aim of the game is to guide the balloon towards the blue rings that appear along the way. The progressive position of the rings is pseudo-random so that there is a continuous shift. An in-game scene is depicted in Figure 9 and Table 7 contains the list of the extracted parameters. This activity, like Shelf Cans, has only one level. This game is for postural balance and correction reactions, shifting, and attention.

## 5. Results

Each patient according to their signs and symptoms, related to MS, receives a personalized rehabilitation path. The IoMT ReMoVES system on which the STORMS project is based favors the continuity of care after hospitalization. It includes motor and cognitive exergames that incorporate enjoyment, technology, and health care. Indeed, the platform will process all the data to make them clear and available within the Therapist Client, processed as a set of graphs and values. This type of layout provides analysis from a cognitive and rehabilitation perspective and allows the multidimensional team to more easily interpret the information. The analysis of indicators containing features concerning attention, working memory, speed of processing information about the environment or situation simulated by the serious games and those concerning the movements performed to reach the goal, defines a complete picture of the physical and mental state of the patient allowing then to monitor more accurately the course of symptoms highlighted by the disease.

As the present study is still ongoing, the game sessions and data collected refer to a preliminary experimentation phase performed on a group of frail elders, affected by neurological sequelae of stroke, post-surgery recovery following hip and limb fractures, and short-term complications (injuries, traumas). The generic indicators presented in this Section prove the feasibility and reliability of the approach on the given user elderly group and will be extended to the pwMS STORMS target group. A series of test session graphs are provided in order to present the visual approach that makes signal analysis possible.

The main indicators and graphs are analyzed from a cognitive and motor point of view. In fact, as already pointed out in the previous sections, although the focal point of the STORMS project is cognitive rehabilitation, it is also necessary to study the movement performed by the patient, predict any errors or compensations, as well as analyze the precision and speed of the gesture itself.

### 5.1. Analysis of Cognitive Tasks

Cognitive functions are continually used to accomplish each daily task. A deficit in one or more of these functions results in a range of difficulties in ADLs, work, and social life. The MMSE will be used to assess the state of impairment and/or deterioration of cognitive efficiency. In previous works, the results obtained with MMSE test have been correlated with the data collected by the system and a significant match has been obtained, as shown in Figure 10 with regards to Owl Nest exergame. The average MMSE score of the population was 25.15±1.84. This analysis shows that patients with better cognitive abilities can learn the techniques and strategies to play faster, regardless of their physical conditions: they could benefit from subsequent levels of difficulty in the game.

The same test will be used to ascertain the capacity of cognitive functions of pwMS and a positive correlation between the two methods of investigation is expected.

Moreover, a preliminary analysis concerns the learning curves of the patients according to the obtained game score. Figure 11 shows an example of the learning curve obtained from previous study with elderly and frail patients when playing Owl Nest exergame. A low score is expected in the initial sessions and a subsequent steady improvement during the final sessions.

Similarly, a graph relatives to patients with MS is expected to have the same trends demonstrating how the execution of exergames involving memory, attention, perception, and reasoning lead to an improvement in performance compared to the initial session highlighting, at the same time, that the various programmed objectives are really treated by the STORMS system.

### 5.2. Analysis of Upper-Limb and Trunk Movement

The tool, already widely used, to measure the disability status of the MS patient will be the EDSS. As already mentioned, the inclusion criterion considered will include patients with a score less than 6. The expected trends will be learning curves that will include information related to the motor performance obtained in multiple game sessions. In particular, for each exergame, the proper joints are tracked in order to detect and evaluate the patient movement patterns.

Then, the raw signals are collected, and a series of preprocessing methods, i.e., filtering and segmentation operations (Figure 12), are used to process them and extract the most important characteristics related with the range of motion and the angles.

Referring to the upper-limb movement, the possible indicators are the range of motion (ROM) of the shoulder, and elbow defined as
(1)$$ROM\left(\theta_{shoulder}\right)=\max\theta_{shoulder}-\min\theta_{shoulder},$$
(2)$$ROM\left(\theta_{elbow}\right)=\max\theta_{elbow}-\min\theta_{elbow},$$
where the angles are described by the following equations:(3)$$\theta_{shoulder}=\arctan\frac{z_{elbow}-z_{shoulder}}{x_{elbow}-x_{shoulder}},$$
(4)$$\theta_{elbow}=\theta_{shoulder}-\arctan\frac{z_{elbow}-z_{wrist}}{x_{elbow}-x_{wrist}}.$$

Possible compensations with the trunk can be observed using the following formula:(5)$$ROM\left(\theta_{trunk}\right)=\max\theta_{trunk}-\min\theta_{trunk},$$
where
(6)$$\theta_{trunk}=\arctan\frac{z_{spine\_shoulder}-z_{spine\_middle}}{y_{spine\_shoulder}-y_{spine\_middle}}.$$

Last, on the basis of the exergames considered, the precision of the movement and the trajectory will be studied. The optimal trajectory and the trajectory performed by the patient will be computed and compared. A small angle between the two lines will mean the movement has been controlled and precise.

A preliminary study on Shelf Cans will be made calculating the average distance between the position of the hand (N samples over time) and the shortest line passing through the origin (location where the tin cans appear) and the targets (the correct matching shelf):(7)$$d=\frac{1}{N}\sum_{n=1}^{N}\frac{\left|\left(y_{t}-y_{o}\right)x_{n}-\left(x_{t}-x_{o}\right)y_{n}+x_{t}y_{o}-y_{t}x_{o}\right|}{\sqrt{{\left(y_{t}-y_{o}\right)}^{2}+{\left(x_{t}-x_{o}\right)}^{2}}}$$
where $hand_{n}=\left(x_{n},y_{n}\right)$ (*x* and *y* are the horizontal/vertical coordinates of the coronal plane); $origin=(x_{o},y_{o})$; $target=(x_{t},y_{t})$. Lower values of *d* are better because they indicate a more precise path.

In previous works, studies relative the evolution over time of the performance of an elderly patient affected from fibula and ischio-pubic fractures is conducted. Figure 13 shows a motor-cognitive analysis when the patient was playing Shelf Cans activity. Despite the trajectories seeming more controlled and precise, the patient hesitates and fails to lead the can of soda towards the shelf of the corresponding color, indicating a cognitive deficit. The average distance *d* is 1.72 which is a quite large value as compared to a healthy population.

More in-depth studies concerning MS patient will be discussed after the collection of their game sessions.

## 6. Conclusions

In the current study, the use of the IoT system ReMoVES as a support to MS rehabilitative treatment and the related monitoring is disclosed. Patients’ activity can be tracked even when performed without the therapist supervision by means of low-cost off-the-shelf components and an easy-to-use interface. This also allows the patients to follow their personal plan of care with continuity also at their home. The current discussion addresses several aspects that are tackled by the presented solution and that derive from its application.

**Patient management point of view:** The use of IoMT devices favors the intensive and continuous practice of rehabilitation and task-oriented exercises for MS patients. The resulting cognitive improvement may lead to a better adaptation to the intrinsic difficulties of the disease. The playful component of exergame can be helpful in increasing motivation and adherence to rehabilitation. At the same time, the muscle strengthening that derives from the practice of these exercises favors the improvement of the patients’ functionality, in order to make them as self-sufficient as possible in carrying out daily activities. Overall, these outcomes can allow a better management of the patients who will not feel abandoned once discharged from the healthcare facilities, but can be continuously monitored and supported by the clinical staff even during home practice.

**Social point of view:** MS patients, as well as patients suffering from other disabling pathologies, are in need of particular attention and care in order to face the everyday challenges deriving from their conditions. Here, IoMT and Digital Technology are crucial for promoting adaptation and coexistence with the disease. The Covid-19 pandemic has highlighted the huge potential relying in medical technologies and now it is mandatory to leverage on them in the everyday life, to address the several issues of other diseases and improve the lives of patients. It is therefore necessary that institutions promote the realization of strategic plans to bring novel technologies to everyone. In such a context, solutions like the one here presented should be replicated on large scale and forwarded to other applications.

**Theoretical point of view:** IoMT solutions enable data collection which improves health diagnoses and patients monitoring. To this purpose, the extraction of meaningful features, along with the use of signal and data processing operations can provide the clinical staff with a clear picture of patients’ conditions. The main implication in such a context is the development of data mining and artificial intelligence techniques for supporting clinical practice. This is a task that will be tackled in the following phases of the project STORMS, after having visualized and made acquainted with the collected data.

**Future work:** In the next months, the experimental phase of the project will take part. An operational protocol will be defined and tested along with the exergames, first in hospital and then at home.

Meanwhile, data analytics will be implemented and strengthened more thoroughly.

As a further development of the project, the activity of ReMoVES could be combined with other IoT services or additional devices for the evaluation of brain functions, for example electroencephalography (EEG), or stimulation (for example, transcranial direct current stimulation or tDCS), in order to better monitor the patient’s practice.

As already suggested in the *social-point-of-view* section, the application of this project can be exported to other diseases, to design an aid framework based on IoMT devices to address the different needs of different diseases.

Indeed, the STORMS project was thought to act as a groundwork for the development of digital telerehabilitation solutions that support MS patients and others in order to improve their quality of life and help them in the activity of daily living.

Finally, ReMoVES system is under continuous updating, it can integrate other sensors and IoT systems, and is easily adapted to new technological devices as, for instance, the new Azure Kinect instead of the current Kinect v2.

## Figures and Tables

**Figure 1 sensors-21-08436-f001:**
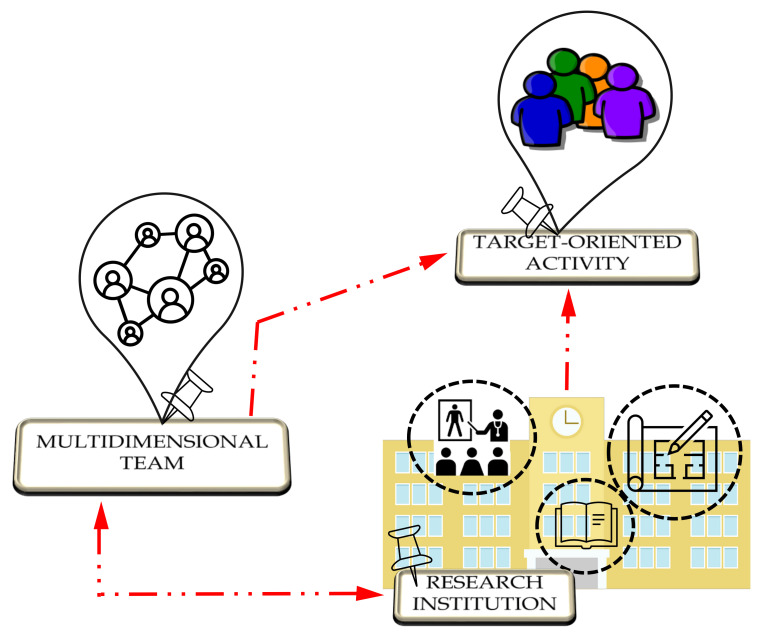
Developers viewpoint architecture.

**Figure 2 sensors-21-08436-f002:**
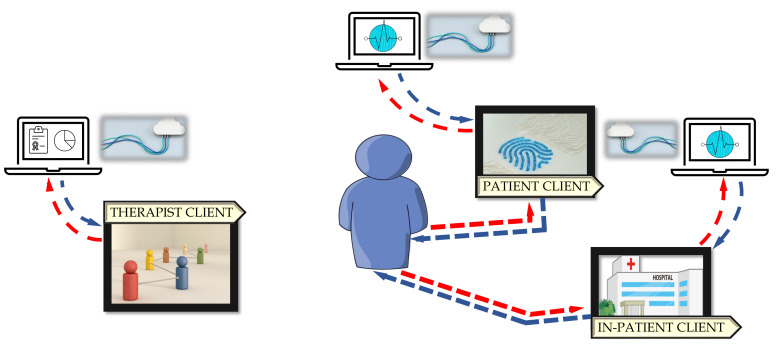
Patient viewpoint architecture.

**Figure 3 sensors-21-08436-f003:**
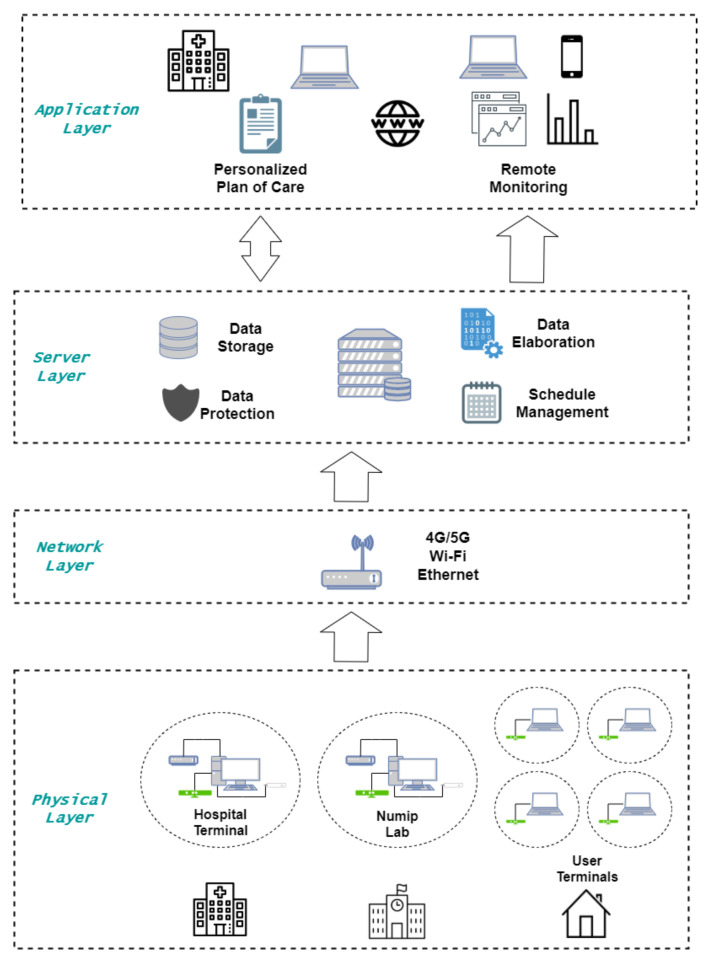
IoT architecture of ReMoVES system.

**Figure 4 sensors-21-08436-f004:**
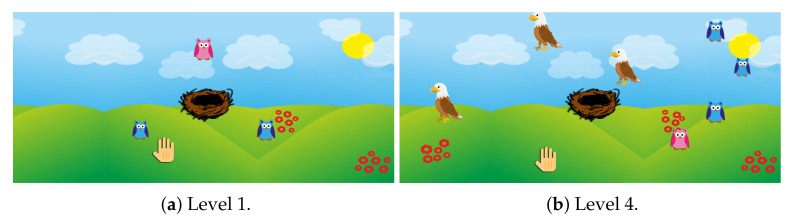
Screenshots of some levels of Owl Nest exergame.

**Figure 5 sensors-21-08436-f005:**
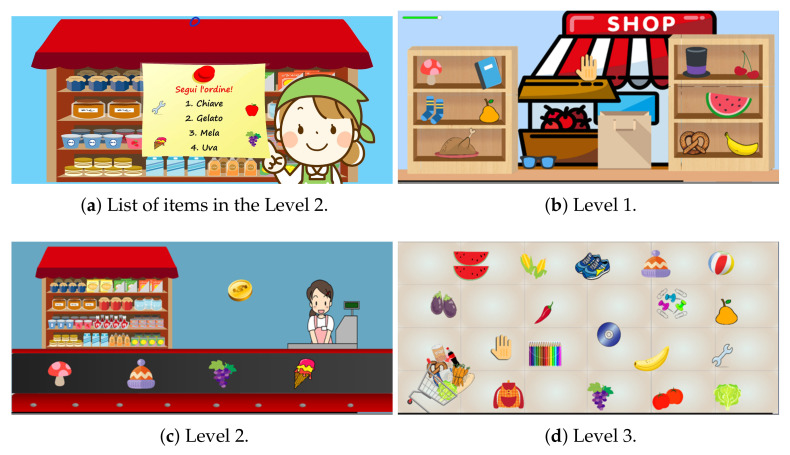
Some screenshots of the levels of Supermarket exergame.

**Figure 6 sensors-21-08436-f006:**
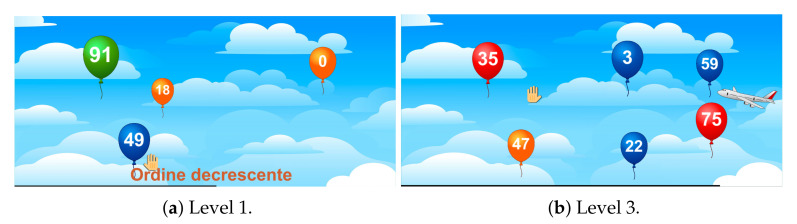
Screenshots of the level one and level three of Numbers exergame.

**Figure 7 sensors-21-08436-f007:**
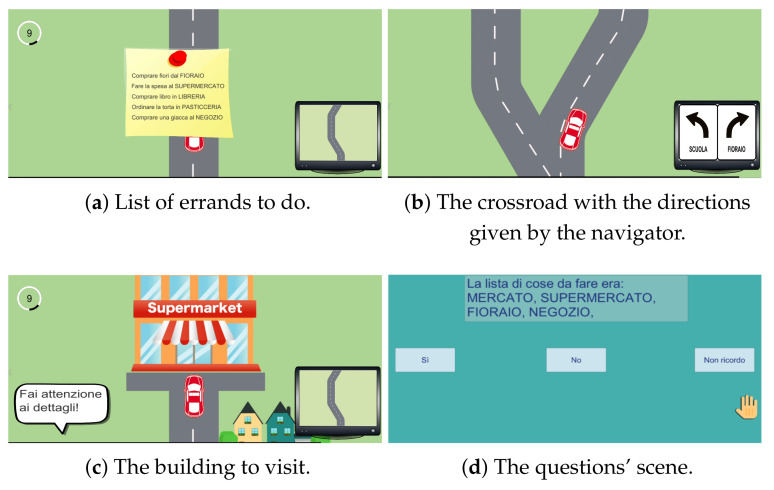
Screenshots of the Business By Car exergame.

**Figure 8 sensors-21-08436-f008:**
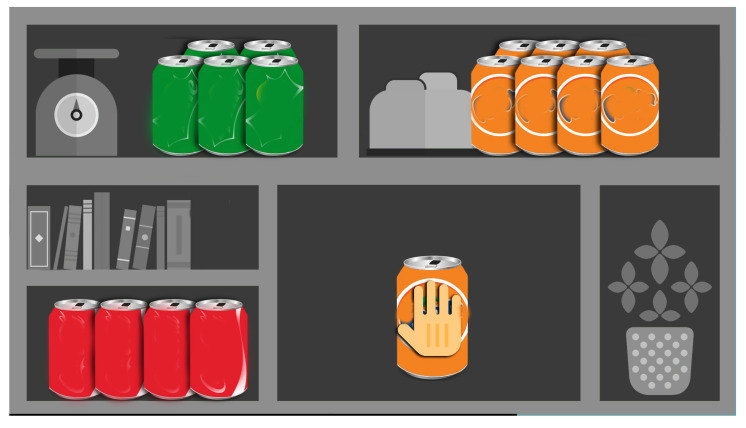
Screenshot of Shelf Cans exergame.

**Figure 9 sensors-21-08436-f009:**
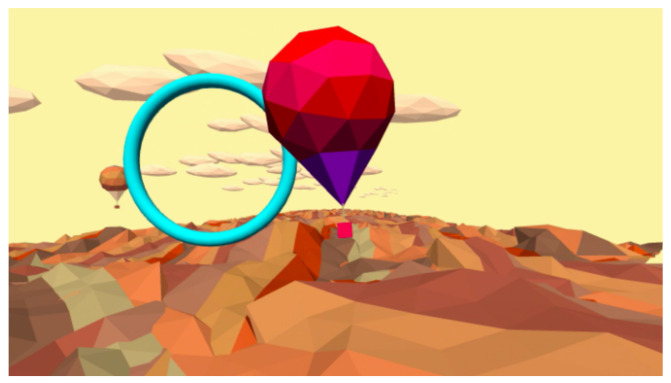
Screenshot of Hot Air exergame.

**Figure 10 sensors-21-08436-f010:**
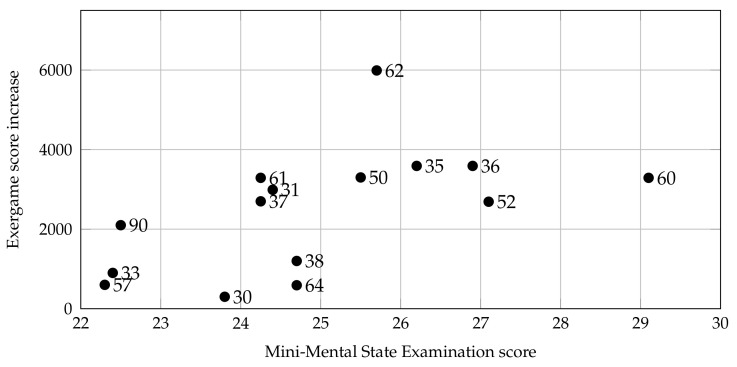
Scatter plot of the MMSE score versus the maximum in-game score increase achieved by the patients, calculated as the difference between their overall best result and their first session. Each point is labeled with patient ID.

**Figure 11 sensors-21-08436-f011:**
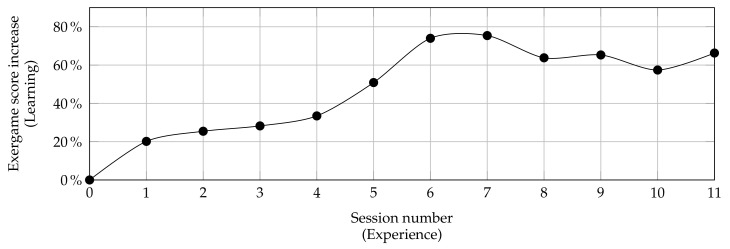
Learning curve of frail elderly population in the context of Owl Nest activity. On the vertical axis, it is evaluated how gaming performance improves with more experience (number of sessions). The game performance is defined as a percentage increase compared to the initial session.

**Figure 12 sensors-21-08436-f012:**
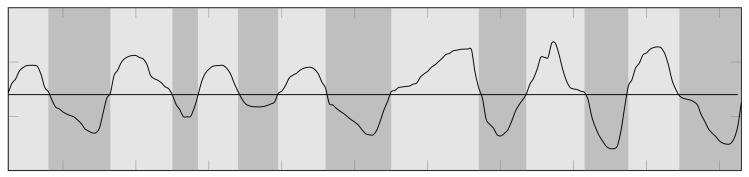
Example of movement segmentation in Hot Air exergame. Dark gray areas are for right displacement, while light gray areas are for the left displacement.

**Figure 13 sensors-21-08436-f013:**
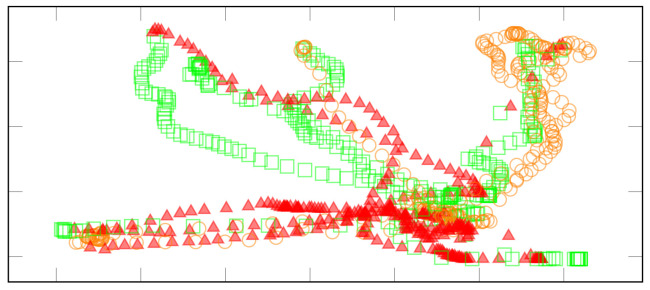
Each point corresponds to the temporal position of the tin can on the screen. Orange circles, red triangles, and green squares refer to orange cans, red cans, and green cans, respectively. In the session, the subject has a controlled movement but sometimes delivers the tin can with hesitation in the wrong shelf as defined by the high value distance *d*.

**Table 1 sensors-21-08436-t001:** Targeted cognitive domains of each exercise.

Games	Targeted Exercise Gains
Owl Nest	- Hand–eye coordination of voluntary arm movements - Reaction time - Inhibition and selective attention - Processing speed
Supermarket	- Hand–eye coordination of voluntary arm movements - Working memory - Visual memory - Inhibition and selective attention - Processing speed
Numbers	- Coordination of voluntary limb movements - Reaction time - Inhibition and selective attention - Processing speed - Hand-eye coordination - Shifting
Business By Car	- Visual memory - Attention - Weight shifting - Postural balance and correction reactions
Shelf Cans	- Hand–eye coordination of voluntary arm movements - Reaction time - Inhibition and selective attention - Processing speed
Hot Air	- Weight shifting - Postural balance and correction reactions - Shifting - Attention

**Table 2 sensors-21-08436-t002:** Parameters extracted in Owl Nest exergame.

Features	Description
CorrectTarget	Number of owls correctely grabbed
WrongTargetEagle	Number of eagles grabbed
WrongTargetBlueOwl	Number of blue owls grabbed
PinkOwlMissed	Number of pink owls missed
OwlPositionX-Y	x and y positions of the grabbed owl
NestPositionX-Y	x and y positions of the nest
TotalOwl	Total number of spawned owl
TotalEagle	Total number of spawned eagle
OwlIDSpawn	ID of the spawned owl
EagleIDSpawn	ID of the spawned eagle
IDgrabbed	ID of the grabbed owl
pinkOwlMissID	ID of the missed pink owl

**Table 3 sensors-21-08436-t003:** Parameters extracted in Supermarket exergame.

Features	Description
CorrectTarget	Number of target correctly grabbed
WrongTarget	Number of target erroneously grabbed
CorrectTargetNOList	Number of target not in the list grabbed but semantically correct (only level 1)
ObjOutOfSequence	Number of target grabbed out of sequence (only level 2, 3, 4)
PassedLevel	Number of completed round
TargetPositionX-Y	x and y positions of the grabbed target
BagPositionX-Y	x and y positions of the shopping bag (only level 1)
Hand-Object	if the hand has grabbed an object (0 for no object grabbed, 1 for object grabbed)

**Table 4 sensors-21-08436-t004:** Parameters extracted in Numbers exergame.

Features	Description
Round	Number of total round
Text	Text visualized on screen for each round
CorrectRound	Number of successfully completed rounds
WrongRound	Number of incorrectly completed rounds
BalloonSpawned	Number of balloons displayed on screen
ColorBalloon	Colors of the current spawned balloons
SpawnPosition	Positions of the current spawned balloons
ScaleBalloon	Sizes of the current spawned balloons

**Table 5 sensors-21-08436-t005:** Parameters extracted in Business By Car exergame.

Features	Description
Scene	The current active scene (1 for game scene, 2 for questions scene)
Speed	The car’s speed
OutOfPath	Number of times the car has gone off the path
ForcedRestart	Number of times the car has been re-positioned on the road
WrongInternalPath	Number of times the car has passed on the path in the middle of the crossroads
ElementList	The elements shown on the list
CorrectPath	Number of times the patient has taken the correct path at the crossroads
WrongPath	Number of times the patient has taken the wrong path at the crossroads
CounterShop	Number of buildings on the list reached
Stop	Indicate if the user has stopped in front of the building (0 if he/she did not stop, 1 if he/she stopped)
Signboard	0 if the correct path was at left or 1 if the correct path was at right
TotQuestions	total number of questions
CorrectAnswer	number of correct answers
WrongAnswer	number of wrong answers
NumQuestions	number of the current question
ActualQuestion	number of questions asked so far
CorrectButton	button where there was the correct answer
AnswerButton	button pressed by the user

**Table 6 sensors-21-08436-t006:** Parameters extracted in Shelf Cans exergame.

Features	Description
CorrectTarget	Number of cans correctly placed
WrongTarget	Number of cans misplaced
originPositionX-Y	x and y positions of the site where the cans appear
targetPositionX-Y	x and y positions of the target site

**Table 7 sensors-21-08436-t007:** Parameters extracted in Hot Air exergame.

Features	Description
CorrectTarget	Number of rings correctly taken
WrongTarget	Number of missed rings

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
