# Peer review of "A Study Protocol for Occupational Rehabilitation in Multiple Sclerosis"

_sensors, 2021, doi:10.3390/s21248436_

Round 1

Reviewer 1 Report

Dear authors, 

I appreciate your work and have only minor points to implement: 

IoMT   do not use abbreviations as key words

I would recommend to add occupational rehabilitation into key words

Can you add the number of the project registration?

I would recommend to add into the heading an information about the style of the article (e.g. study protocol).

Author Response

We would like to thank the reviewers for their valuable comments. Here, we reply to their concerns and underline the modifications we performed, according to their suggestions.

In general, the Introduction, the State of the Art, the Material and Methods and the Results Sections have been deeply revised and partially  rewritten.

In the other sections, novel or modified parts are written in red, in order to help reviewers for further readings.

In Section 2, the originality of ReMoVES system are now better presented and the literature background on already existing solutions has been consolidated and compared.

In Section 3 the architecture of ReMoVES, the operational protocol and the functioning of the patient client is added.

The presentation of the activities involve in the project STORMS are now discussed in Section 4, and possible methods for analysis and evaluation of the results is presented in the new Section 5 - Results.

A punctual response to reviewers’ comments follows.

  • IoMT do not use abbreviations as key words; I would recommend to add occupational rehabilitation into key words.

The suggested keywords are modified or added.

  • Can you add the number of the project registration?

The project STORMS won the "2020 Innovazione Digitale nella Sclerosi Multipla" award, sponsored by Merck; no registration number has been assigned.

  • I would recommend to add into the heading an information about the style of the article (e.g. study protocol).

The Title of the paper was modified as suggested. In the Abstract, a sentence is added in order to better explain the style of the article.

Reviewer 2 Report

The authors describe the STORMS project, more specifically the exergames that were realized for MS patients.

Though I have a lot of appreciation for this kind of efforts, investigating the best possible strategies to support and rehabilitate (MS) patients, I am not impressed by the current paper. Summarized in one sentence, the reason why I have doubts is the lack of research contribution in the article. It shortly mentions the overall project context, and mainly contains a feature description of the exergames. It cannot be considered an engineering paper, as detailed information on e.g. the engineering process and the software architecture and technology is missing. On the other hand, it is not a “design paper” either, as there is no thorough description of the (theoretical) foundation for the game concepts and supported skill training. Also, no evaluation is described in this stage. Therefore, there is no clear research contribution that would trigger the reader to read the paper. This type of exergames for MS exists for many years, in research context as well as some commercial rehabilitation setups (e.g. with robot or sensor-based support, and indeed with camerabased movement detection such as with Kinect).

Another general remark I have, relates to the terminology IoT and IoMT. Typically these systems consist of small components that exchange info via the Internet, and that behave as intelligent components in the overall system. Though the authors refer at some point to “wearable devices”, it is not clear at all if there is a link between IoT and the system setup used in the exergames, with a Kinect to detect movements. The overall system is a telerehabilitation system, with internet applications, but that - for what we see in this paper -  is of no interest for researchers looking for true IoT examples. It is also not clear to what extent the underlying ReMoVES system contains IoT support that is not used in the presented exergames.

The introduction is very long and could be structured better. It is hard to discover the natural storyline, as in some places it seems like a collection of unconnected paragraphs on separate topics. This is particularly the case where “exergames” are introduces, and it is not clear whether the authors intend to situate the type of exergames or just want to position it within other solutions such as VR with other input modalities. In the same way there is a strange transition / story line from VR / IoT to telerehabilitation, which basically are things on different conceptual levels.

The authors do not succeed in highlighting what is the real contribution and what is different from other existing work, neither in the introduction nor in the related work section (called “Existing solutions and differences”). This section is too short and lacks a clear positioning within possible approaches, it is just an enumeration of some systems without even clearly identifying what the scope is of the related work section (gamification? Camerabased systems? IoT?).

We collaborated with several rehabilitation centers who use exergames for MS patients (both for cognitive as for motor training) in daily rehab practice, so the statement that these systems are not yet out there is wrong. There is a lot of research, and several commercial systems. The concluding sentences of section 2 on ReMoVES cannot convince at all why the presented games / rehab approach is innovative.

Section 3, Materials and Methods, is als not structured in a smooth way. For instance the first subsection on ReMoVES is “sloppy”, and the section on “inclusion” is very weird. One would describe inclusion criteria for a study, but a study is not presented in the article. The target audience could be described indeed.

It is really weird that in the beginning of section 4, Results, the focus of the games seems to be cognitive training whereas the previous and following sections pay more attention to motor skill training and movements in the exergames. Twice it is mentioned that the games are adaptive to the user(‘s performance) but it is not explained how exactly.

With all respect for the really nice game concepts, but it makes no sense to describe them in this way with a table for collected data. We indeed have no insight in data collected and analysed, so the extensive tables are not useful. Similarly, despite the nice games, the extensive description of all of them could be replaced by a small number of example games/concepts.

The discussion is not really a discussion, but just a section with loosely connected paragraphs on “what will be done”. A reader of an article searchers for what has been done, in this case how the game concepts and implementations lead to research results (next project stages).

A lot of research has been done in this field by many international research groups. However, the number of "Italian" references is excessively high.

Some of these are self references (number seems acceptable), but the other references could reflect better the international community involved.

Author Response

We would like to thank the reviewers for their valuable comments. Here, we reply to their concerns and underline the modifications we performed, according to their suggestions.

In general, the Introduction, the State of the Art, the Material and Methods and the Results Sections have been deeply revised and partially  rewritten.

In the other sections, novel or modified parts are written in red, in order to help reviewers for further readings.

In Section 2, the originality of ReMoVES system are now better presented and the literature background on already existing solutions has been consolidated and compared.

In Section 3 the architecture of ReMoVES, the operational protocol and the functioning of the patient client is added.

The presentation of the activities involve in the project STORMS are now discussed in Section 4, and possible methods for analysis and evaluation of the results is presented in the new Section 5 - Results.

A punctual response to reviewers’ comments follows.

  • It cannot be considered an engineering paper, as detailed information on e.g. the engineering process and the software architecture and technology is missing. Another general remark I have, relates to the terminology IoT and IoMT. Typically these systems consist of small components that exchange info via the Internet, and that behave as intelligent components in the overall system. Though the authors refer at some point to “wearable devices”, it is not clear at all if there is a link between IoT and the system setup used in the exergames, with a Kinect to detect movements. The overall system is a telerehabilitation system, with internet applications, but that - for what we see in this paper -  is of no interest for researchers looking for true IoT examples. It is also not clear to what extent the underlying ReMoVES system contains IoT support that is not used in the presented exergames.

As said in the paper, the architecture of IoT ReMoVES system is described in detail in the past article [29] and is here biefly summarized. In Section 3 the architecture of the users and developers viewpoints (as suggested in IEEE standards [54] ) are depicted in Fig.1 and 2, respectively. The four-layer general architecture is given along with a corresponding drawing and a brief description.

  •  The introduction is very long and could be structured better. It is hard to discover the natural storyline, as in some places it seems like a collection of unconnected paragraphs on separate topics. This is particularly the case where “exergames” are introduces, and it is not clear whether the authors intend to situate the type of exergames or just want to position it within other solutions such as VR with other input modalities. In the same way there is a strange transition / story line from VR / IoT to telerehabilitation, which basically are things on different conceptual levels.

The Introduction and the SoA, in Sections 1 and 2, have been rewritten and better structured. 

  • The authors do not succeed in highlighting what is the real contribution and what is different from other existing work, neither in the introduction nor in the related work section (called “Existing solutions and differences”). This section is too short and lacks a clear positioning within possible approaches, it is just an enumeration of some systems without even clearly identifying what the scope is of the related work section (gamification? Camera based systems? IoT?). We collaborated with several rehabilitation centers who use exergames for MS patients (both for cognitive as for motor training) in daily rehab practice, so the statement that these systems are not yet out there is wrong. There is a lot of research, and several commercial systems. The concluding sentences of section 2 on ReMoVES cannot convince at all why the presented games / rehab approach is innovative. 

Literature review has been improved and the whole Section 2 is modified to make a better comparison with existing solutions and highlights the differences.

  • Section 3, Materials and Methods, is also not structured in a smooth way. For instance the first subsection on ReMoVES is “sloppy”, and the section on “inclusion” is very weird. One would describe inclusion criteria for a study, but a study is not presented in the article. The target audience could be described indeed.

 The description of ReMoVES was improved. Also, in Section 3 other paragraphs were added to explain in details the operative protocols.

  •  It is really weird that in the beginning of section 4, Results, the focus of the games seems to be cognitive training whereas the previous and following sections pay more attention to motor skill training and movements in the exergames. Twice it is mentioned that the games are adaptive to the user(‘s performance) but it is not explained how exactly. 

We modified some sentences in order to better explain the focus of the games.

  •  With all respect for the really nice game concepts, but it makes no sense to describe them in this way with a table for collected data. We indeed have no insight in data collected and analysed, so the extensive tables are not useful. Similarly, despite the nice games, the extensive description of all of them could be replaced by a small number of example games/concepts.

 The tables have been shortened, eliminating the features in common with other games. These common features are explained in the paragraph 4.1.

  • The discussion is not really a discussion, but just a section with loosely connected paragraphs on “what will be done”. A reader of an article searchers for what has been done, in this case how the game concepts and implementations lead to research results (next project stages). 

The Results Section has been added along with a discussion relatively the experimental phase carried out on a generic elderly group.

Section 6 becomes Conclusions.

  • A lot of research has been done in this field by many international research groups. However, the number of "Italian" references is excessively high. Some of these are self references (number seems acceptable), but the other references could reflect better the international community involved. 

We added other references to reflect better the whole international community. 

Reviewer 3 Report

The authors describe the STORMS project that implements a rehabilitative protocol of exercises based on ReMoVES services that has already been published in the Sensors. The main outcome of this work are the six exergames that have been developed. To this end, the work seems valuable. However, relevant systems have been developed and proposed in the literature. Furthermore, these systems have been extensively evaluated. This point seems to be one the study's drwabacks.

Also, the authors do not provide any justification on the platform's low cost features.

A comparison with commercial products e.g. Jintronix would be possible.

Finally, as kinect is discontinued, can ReMoVES be adapted to other peripherals (e.g. kinect Azure)?

Author Response

We would like to thank the reviewers for their valuable comments. Here, we reply to their concerns and underline the modifications we performed, according to their suggestions.

In general, the Introduction, the State of the Art, the Material and Methods and the Results Sections have been deeply revised and partially  rewritten.

In the other sections, novel or modified parts are written in red, in order to help reviewers for further readings.

In Section 2, the originality of ReMoVES system are now better presented and the literature background on already existing solutions has been consolidated and compared.

In Section 3 the architecture of ReMoVES, the operational protocol and the functioning of the patient client is added.

The presentation of the activities involve in the project STORMS are now discussed in Section 4, and possible methods for analysis and evaluation of the results is presented in the new Section 5 - Results.

A punctual response to reviewers’ comments follows.

  • The main outcome of this work are the six exergames that have been developed. To this end, the work seems valuable. However, relevant systems have been developed and proposed in the literature. Furthermore, these systems have been extensively evaluated. This point seems to be one the study's drwabacks. A comparison with commercial products e.g. Jintronix would be possible. 

Literature review has been improved and the whole Section 2 is modified to make a better comparison between existing solutions and highlights the differences.

  • Finally, as kinect is discontinued, can ReMoVES be adapted to other peripherals (e.g. kinect Azure)?

 The next objective is the updating of the technological devices of ReMoVES to improve the potential and the performance of the system. For instance, the current exergames could be adapted for the new Azure Kinect sensor.

Reviewer 4 Report

The present paper describes the STORMS (Solution Towards Occupational Rehabilitation in Multiple Sclerosis) project and its preliminary phase which consists on designing exergames for supporting cognitive and motor rehabilitation in patients with neurological disability. The designed exergames are adequate and good contextualized for facilitating the ADL in the panorama of occupational rehabilitation for Sclerosis Multiple patients.

The paper is well written and structured with an interesting approach. However, despite the fact that it is said to be a preliminary phase, in order to improve the technical scientific impact, it would be highly recommended to include results of its use with patients and a discussion of the results obtained in the proposed rehabilitation activities.

Therefore, in the case that the authors provide some implementation results on a sample of MS patients, this work could be accepted to be published in Sensors journal.

Some minor types or recommendations:

  • Line 116: “The” instead of “the”
  • The sections 4.6 and 4.7 could describe the three different levels that patiens can evolute as the rest of exergames described.
  • Section 4 should be title developed tools for occupational rehabilitation (or similar) and a new section for results with patients is expected.

Author Response

We would like to thank the reviewers for their valuable comments. Here, we reply to their concerns and underline the modifications we performed, according to their suggestions.

In general, the Introduction, the State of the Art, the Material and Methods and the Results Sections have been deeply revised and partially  rewritten.

In the other sections, novel or modified parts are written in red, in order to help reviewers for further readings.

In Section 2, the originality of ReMoVES system are now better presented and the literature background on already existing solutions has been consolidated and compared.

In Section 3 the architecture of ReMoVES, the operational protocol and the functioning of the patient client is added.

The presentation of the activities involve in the project STORMS are now discussed in Section 4, and possible methods for analysis and evaluation of the results is presented in the new Section 5 - Results.

A punctual response to reviewers’ comments follows.

  • The paper is well written and structured with an interesting approach. However, despite the fact that it is said to be a preliminary phase, in order to improve the technical scientific impact, it would be highly recommended to include results of its use with patients and a discussion of the results obtained in the proposed rehabilitation activities.

Since this study is a work in progress and a study protocol, MS patients have not been recruited already. An experimental phase has been carried out on a generic elderly group as described at Section 5, with possible analysis and indicators which prove the feasibility and reliability of the approach.

  • The sections 4.6 and 4.7 could describe the three different levels that patiens can evolute as the rest of exergames described.

 The activities presented in subsections 4.6 and 4.7 have only one level. We have specified it better at the beginning of Section 4 and in their description.

  • Section 4 should be title developed tools for occupational rehabilitation (or similar) and a new section for results with patients is expected.

The title of Section 4 was modified as the Reviewer suggested.

Round 2

Reviewer 3 Report

My remarks have been considered and explained in the revised version.

Reviewer 4 Report

Despite the partially changes made by the authors, they do not really respond to the arguments mentioned by the reviewers. Therefore, as currently paper’s state, it is not considered ready for publication in this journal.